# A Novel Variant of the *ACTRT1* Gene Is Potentially Associated with Oligoasthenoteratozoospermia, Acrosome Detachment, and Fertilization Failure

**DOI:** 10.3390/genes16121422

**Published:** 2025-11-28

**Authors:** Olga Solovova, Sabina Khayat, Sergey Bogolyubov, Elizaveta Bragina, Tatiana Cherevatova, Vyacheslav Chernykh

**Affiliations:** 1Research Centre for Medical Genetics, 115522 Moscow, Russia; olga_pilyaeva@list.ru (O.S.); sabina_hayat@mail.ru (S.K.); bragor@mail.ru (E.B.); tatiana_milovidova@mail.ru (T.C.); 2Endocrinology Research Centre, 117292 Moscow, Russia; sergej_bogolyubo@mail.ru; 3Next Generation Clinic, 107140 Moscow, Russia; 4Belozersky Institute of Physico-Chemical Biology, Lomonosov Moscow State University, 119992 Moscow, Russia; 5Institute of Biomedicine, Pirogov Russian National Research Medical University, 117997 Moscow, Russia

**Keywords:** perinuclear theca, acrosome, TEM, spermatozoa, *ACTRT1*, oligozoospermia, teratozoospermia, male infertility, whole exome sequencing

## Abstract

**Background:** Male infertility is a common reproductive disorder, affecting about 7% of men in the general population. Despite its prevalence, the cause of infertility is often unknown. This case report presents the results of a comprehensive evaluation of a patient with severe oligoasthenoteratozoospermia and primary infertility. **Methods:** The patient underwent clinical, andrological, and genetic examinations, including semen analysis, transmission electron microscopy, cytogenetic examination, molecular analysis of the AZF locus and the *CFTR* gene, whole-exome sequencing, and Sanger sequencing. **Results:** Semen analysis revealed severe oligoasthenoteratozoospermia. Transmission electron microscopy showed acrosome detachment from the nucleus in 49% of the spermatozoa. A high percentage (54%) of spermatozoa with insufficiently condensed (“immature”) chromatin was also observed. No chromosomal abnormalities, Y chromosome microdeletions, or pathogenic *CFTR* gene variants were identified. Whole-exome sequencing revealed a novel c.821G>C variant (chrX:127185365G>C; NM_138289.4) in the *ACTRT1* gene (Xq25). This variant was hemizygous in the patient and heterozygous in his mother, as determined by Sanger sequencing. According to the ACMG guidelines (PM2, PP3), this missense variant in the *ACTRT1* gene was classified as a variant of uncertain clinical significance (VUS). Amino acid conservation and 3D protein modeling predict that the identified variant has a deleterious effect on the protein. **Conclusions:** This study suggests a potential link between a novel *ACTRT1* variant and a specific teratozoospermia phenotype. Further functional studies are needed to confirm this association and determine the role of the gene in X-linked male infertility.

## 1. Introduction

Male infertility affects around 7% of men worldwide and contributes to nearly half of all infertility cases in couples [1]. Despite advances in molecular genetics, the cause of azoospermia, severe oligozoospermia, and rare syndromic forms of astheno-/teratozoospermia remains unclear for many patients [2]. Many idiopathic cases are attributed to disturbances during the final phase of spermatogenesis, known as spermiogenesis, wherein round spermatids undergo extensive remodeling to form mature spermatozoa [3]. This process involves a series of highly coordinated morphological changes, such as nuclear elongation, chromatin condensation, acrosome biogenesis, the development of the flagellar axoneme and the periaxonemal structures, the assembly of the mitochondrial sheath in the midpiece, and the extrusion of residual cytoplasm [3]. These changes are critical for motility and fertilization competence [4].

The perinuclear theca (PT), a major cytoskeletal component of the sperm head, consists of various cytosolic and nuclear proteins [5]. The PT is conventionally divided into two functionally distinct regions: the subacrosomal region (SAR), located between the inner acrosomal membrane and the nuclear envelope, and the postacrosomal region (PAR), positioned between the plasma membrane and the nuclear envelope [6,7]. PT-SAR proteins are derived from acrosomal vesicles during acrosome biogenesis, whereas PT-PAR proteins are synthesized in the cytoplasmic lobe and transported across the manchette [8]. Thus, one of the critical functions of PT-SAR proteins is to anchor the developing acrosomes to the sperm nucleus, whereas PT-PAR proteins (e.g., phospholipase C zeta, PLCζ) are primarily involved in the later stages of the fertilization process, such as oocyte activation [5,6,7,9].

ACTRT1 (Actin-Related Protein Testis 1), which is encoded by a single-exon *ACTRT1* gene (OMIM: * 300487) located on the X chromosome (Xq25), is a vital component of the PT-SAR-specific actin-related protein complex (ACTRT1-ACTRT2-ACTRT3-ACTL7A-ACTL7B-ACTL9) that anchors developing acrosomes to the nucleus [10]. This complex immunoprecipitates with the inner acrosomal membrane protein SPACA1 and the nuclear envelope proteins PARP11 and SPATA46 [11]. In *Actrt1* knockout mice, the loss of Actrt1 attenuates the interaction between Actl7a and Spaca1 proteins, resulting in severe subfertility characterized by morphologically abnormal sperm heads with detached acrosomes and partial fertilization failure [11]. Similarly, a recent human study identified a 110 kb hemizygous loss-of-function deletion affecting the *ACTRT1* gene in infertile men, presenting with severe oligoteratozoospermia, detached acrosomes, and markedly reduced fertilization rates [12]. As demonstrated by Q. Zhang et al., Actrt1 and Actl7a exhibit a spatiotemporal localization in male germ cells. These proteins are specifically targeted to the SAR in round and elongated spermatids to mediate acrosome–nucleus anchoring during acrosome biogenesis and subsequently relocate to the PAR in mature spermatozoa, positioning them for potential roles in later fertilization stages, such as oocyte activation [11]. Notably, loss or reduced expression and/or abnormal localization of PLCζ has been reported in *ACTRT1*/*ACTL7A*/*ACTL9* mutant patients, as well as in *Actrt1*-, *Actl7a*-, and *Actl9*-deficient mice [11,12,13,14,15,16,17,18,19,20,21,22,23,24]. Furthermore, disruption of *ACTL7A* or *ACTL9* has been consistently shown to cause acrosome detachment, total fertilization failure, and male infertility in both mice and humans [13,14,15,16,17,18,19,20,21,22,23,24].

This report details the identification of a novel *ACTRT1* gene variant in a patient exhibiting a specific teratozoospermia phenotype. We provide a detailed clinical, morphological, and genetic characterization of the case and discuss its potential contribution to understanding the genetic basis of male infertility.

## 2. Materials and Methods

### 2.1. Patient

A 33-year-old male patient (L.A., Ex2862) was referred to the Research Centre for Medical Genetics in 2022 for medical genetic counseling and examination due to primary infertility. His clinical history included recurrent severe oligoasthenoteratozoospermia (sOAT), as well as the surgical treatment of right inguinal cryptorchidism via orchiectomy, which was performed in December 2022. Preoperative scrotal ultrasonography revealed significant testicular asymmetry, with a preserved left testicular volume of 16.5 mL and right testicular hypoplasia of 7.64 mL, as well as complete inguinal ectopia. Additionally, a grade I left-sided varicocele was identified. The physical examination was unremarkable, with anthropometric measurements of 178 cm height and 78 kg weight (BMI 24.6 kg/m^2^). An endocrine evaluation revealed elevated levels of luteinizing hormone (11.50 mIU/mL; reference range: 1.70–8.60), indicating compensated hypogonadism [25]. Normal follicle-stimulating hormone levels (11.30 mIU/mL; reference range: 1.50–12.40) and normal total testosterone levels (6.29 ng/mL; reference range: 2.49–8.36) supported this finding. Standard genetic examination revealed a normal 46, XY karyotype with no Y-chromosome microdeletions. Screening for the 22 most common pathogenic *CFTR* variants was negative. Analysis of the polymorphic CAG repeat region of the androgen receptor gene revealed 21 repeats, which is within normal limits.

The patient’s 32-year-old female partner underwent a complete evaluation that demonstrated normal ovarian reserve parameters and a normal female karyotype (46, XX). The couple underwent an assisted reproductive technology protocol involving intracytoplasmic sperm injection (ICSI) with ejaculated sperm. Ovarian stimulation using a gonadotropin-releasing hormone antagonist protocol resulted in the retrieval of 12 metaphase II oocytes. Despite an optimal ovarian response and sufficient mature oocytes, the cycle was complicated by severe fertilization failure, with only one oocyte demonstrating normal fertilization. According to the standard grading criteria, the resulting embryo had poor morphology (grade 3) and did not implant after transfer. A subsequent in vitro fertilization using an ICSI technique incorporating assisted oocyte activation (AOA) with calcium ionophore is planned due to the fertilization failure observed in order to overcome the suspected oocyte activation deficiency.

Written informed consent was obtained from all examined individuals. The study was approved by the Ethics Committee of the Research Centre for Medical Genetics (protocol code: №4/3; date of approval: 19 April 2021).

### 2.2. Standard Semen Examination

Semen samples were collected by masturbation after 3–5 days of sexual abstinence. The analyzed semen parameters included ejaculate volume, viscosity, and pH; concentration and total sperm count; and motility, vitality, and morphology. Sperm vitality (%), motility (%), morphology (%), and sperm count were determined by light microscopy using a Nikon Ci microscope (Nikon Corp., Tokyo, Japan). A standard semen analysis was performed according to the World Health Organization laboratory guidelines (WHO, 2010), and the semen parameters were interpreted based on the reference values established in these guidelines [26].

### 2.3. Transmission Electron Microscopy

For the transmission electron microscopy (TEM) analysis, semen samples were obtained from the patient and a control group of 24 fertile, normozoospermic men aged 35 years or younger who had fathered a child within the past 12 months. All control samples exhibited normal semen parameters. After liquefaction, samples from the patient and controls were fixed with a 2.5% glutaraldehyde solution in a 0.1 M cacodylate buffer solution (pH 7.2–7.4). The samples were then treated with 1% osmic acid and embedded in epoxy resin. Ultrathin sections were prepared using a Reichert–Jung Ultracut E ultramicrotome (Leica Microsystems, Vienna, Austria) and mounted on copper grids covered with Formvar film. The sections were then contrasted using a 1% aqueous solution of uranyl acetate and a lead citrate solution. The preparations were examined at 80 kV using a JEM-1011 transmission electron microscope (JEOL, Akishima, Japan), which was equipped with an Orius SC1000 W camera (Gatan Inc., Pleasanton, CA, USA). Results for each parameter are expressed as percentages.

### 2.4. Isolation of Genomic DNA

Genomic DNA of the proband and his mother was extracted from peripheral blood leukocytes using a standard Wizard^®^ Genomic DNA Purification Kit (Promega, Madison, WI, USA) according to the manufacturer’s protocol.

### 2.5. Whole-Exome Sequencing

Whole-exome sequencing (WES) was performed on the proband’s DNA sample. Libraries were prepared from fragmented genomic DNA using the KAPA Hyper Prep Kit (Roche, Basel, Switzerland), and target enrichment was carried out using KAPA HyperExome probes (Roche, Basel, Switzerland). Paired-end sequencing (2 × 150 bp) was conducted on an Illumina NextSeq 500 platform (Illumina Inc., San Diego, CA, USA).

The resulting raw data were aligned to the human reference genome (GRCh37/hg19) using the NGSData software. Sequencing achieved a mean coverage of 78×, with 97% of target regions covered at least 10×. Variant calling and annotation were performed according to the standard HGVS nomenclature v.21.1.3. The identified variant was visualized using Integrative Genomics Viewer v.2.17.4 (IGV), and its pathogenicity was assessed in accordance with the guidelines of the American College of Medical Genetics and Genomics (ACMG) [27]. A total of 29,362,240 reads were obtained with an average read length of 2 × 150 bp and an average coverage of 78×. Only 6% of the sequences had a coverage of less than 10%. The region chrX:127,184,959-127,186,264, which includes the *ACTRT1* gene, is covered more than 44×, with an average coverage of 64×.

### 2.6. Sanger Sequencing

To validate the variant identified by WES, exon 1 of the *ACTRT1* gene was amplified from genomic DNA using PCR with the following primers, synthesized by Eurogen (Moscow, Russia): forward 5′ AGAGGTCATGATGGATGCACCA, reverse 5′ TTGGGGATGAGCTGTACCAAGT. PCR was performed in 25 µL PCR reactions containing 1 × PCR buffer, 2 mM MgCl_2_, 0.2 mM of each dNTP, 0.5 mM of each primer, 0.3 U of Taq DNA polymerase (Syntol, Moscow, Russia), and 1 µL of genomic DNA. Cycling was performed using a GeneAmp PCR System 9700 Thermal Cycler (Applied Biosystems Inc., Foster City, CA, USA) with the following cycle program: initial denaturation at 95 °C for 2 min, followed by 35 cycles of 95 °C for 30 s, 64 °C for 30 s, and 72 °C for 30 s, followed by a final extension of 5 min at 72 °C. In total, 5 µL of the PCR products was visualized on 2% agarose gels using ethidium bromide staining and UV light transillumination. Amplicons were purified using an Exonuclease I 20 U/μL/FastAP Thermosensitive Alkaline Phosphatase 1 U/μL (Thermo Fisher Scientific, Waltham, MA, USA) mixture. Purified PCR products were sequenced using the BigDye Terminator Kit v3.1 and Applied Biosystems 3500 DNA Analyzer (Thermo Fisher Scientific, Waltham, MA, USA) according to the manufacturer’s protocol. The result of the sequence data was visualized by Chromas software v.2.6.6 (Technelysium, South Brisbane, QLD, Australia).

### 2.7. In Silico Pathogenicity Prediction and Structural Analysis of Detected Variant

The potential pathogenicity of the identified missense variant was assessed using a comprehensive suite of bioinformatics tools, including SIFT, SIFT4G, PROVEAN, MutationTaster, LRT, FATHMM, DANN, MetaLR, MetaSVM, M-CAP, MutationAssessor, MutPred, MVP, PrimateAI, EIGEN, EIGEN-PC, and DEOGEN2.

The evolutionary conservation of the affected amino acid position was analyzed using the UCSC Genome Browser (https://genome.ucsc.edu/, accessed on 20 June 2024). The impact of the amino acid substitution on protein structure was visualized and analyzed using the Project HOPE web portal (https://www3.cmbi.umcn.nl/hope/, accessed on 17 June 2024) [28].

## 3. Results

### 3.1. Semen Analysis

Repeated semen analyses revealed low sperm concentrations ranging from 0.2 to 2.8 (1.1 ± 1.2) million per milliliter and correspondingly reduced total sperm counts of 3.2 ± 3.5 × 10^6^ per ejaculate in the four examined samples (Appendix A). The sperm motility parameters decreased dramatically. Progressive motility (PR) was low, 0–9% (5.3 ± 3.8%), and total motility (PR + NP) was below the reference range (≥40%) in two samples (26% and 37%). However, it was normal in the last two samples (42% and 51%). Sperm vitality exhibited considerable variability (71.5 ± 21.4%), ranging from severely compromised to normal (45–91%). Severe teratozoospermia was also consistently observed in all examined samples (99–100% abnormal morphology), with only 0.5% ± 0.6% of gametes being morphologically normal. Head defects were the predominant morphological abnormality, present in 65–75% (70.0 ± 4.5%) of gametes. The volume and pH of the ejaculate were within the normal range (2.9 ± 0.4 mL and 7.6 ± 0.1, respectively). The leukocyte concentration was within the normal reference range (0.16 ± 0.11 million/mL), thereby excluding an inflammatory etiology (Appendix A).

### 3.2. TEM Results

TEM revealed significant ultrastructural abnormalities in the patient’s spermatozoa (Figure 1C–E,E1 and Appendix A), but not in the control group (Figure 1A,B, Appendix A).

A comparative ultrastructural analysis of sperm morphology revealed an absence of spermatozoa with intact heads in the patient (0%), in contrast to the control group, in which such cells constituted 5.4 ± 1.7% (range, 4–9%). Quantitative analysis of 100 spermatozoa showed that 27% (27/100) lacked an acrosome entirely. Primary absence was observed in 1% (1/100), while secondary absence, indicative of a premature acrosomal reaction, was observed in 26% (26/100). Of the 73 spermatozoa that retained an acrosomal structure, 67% (49/73) had an enlarged subacrosomal space, and 58% (42/73) had acrosomal hypoplasia. Electron-lucent acrosomal content and an absent postacrosomal segment were found in 21% (15/73) and 8% (6/73) of gametes, respectively. Only 6% of the spermatozoa had normal acrosomes but contained non-condensed chromatin. While the patient’s overall incidence of acrosomal hypoplasia (42/100, or 42%) was within the normal reference range (46.0 ± 11.7%, ranging from 7 to 61%), other critical parameters showed significant deviations. The proportion of spermatozoa with impaired chromatin condensation was markedly higher at 54% compared with 17.0 ± 8.3% (range, 7–29%) in the control group. Similarly, spermatozoa with excessive residual cytoplasm accounted for 31% of the patient’s sample, compared to 6.2 ± 3.9% (range, 1–13%) in the control group. The percentage of spermatozoa with an enlarged subacrosomal space was 49%, compared to a control value of 5.3 ± 3.4% (range, 1–11%). Additionally, the incidence of flagellar axoneme abnormalities was substantially higher in the patient (31%) than in the control group (9.7 ± 7.6%, range 1–30%).

The most pronounced pathological features of teratozoospermia detected in the patient were the complete absence of normally formed sperm heads, characterized by detachment of the inner acrosomal membrane from the sperm nucleus, severe chromatin condensation defects, excessive cytoplasmic retention, and a high percentage of enlarged subacrosomal spaces (Figure 1C–E,E1 and Appendix A).

Due to severe oligoasthenoteratozoospermia and specific ultrastructural defects in the gametes revealed by TEM results, the patient was referred for WES to identify the genetic cause of male infertility.

### 3.3. Molecular Genetic, Segregation, and Bioinformatic Study Results

Initial variant filtering and annotation focused on known genes associated with male infertility revealed no pathogenic, likely pathogenic, or variants of unknown clinical significance in established oligoasthenoteratozoospermia-related genes. Subsequent analysis identified a novel hemizygous missense variant in the *ACTRT1* gene (NM_138289.4), located on the X chromosome (locus Xq25). This variant is characterized by a c.821G>C substitution that results in an amino acid change from glycine to alanine at position 274 (Figure 2). A comprehensive review of the Online Mendelian Inheritance in Man (OMIM) database revealed no established associations between the *ACTRT1* gene and male infertility. This suggests that this finding may represent a novel genotype–phenotype correlation. The identified nucleotide sequence variant is not registered in the control sample Genome Aggregation Database (gnomAD v2.1.1) (http://gnomad.broadinstitute.org/ accessed on 7 February 2025).

Evolutionary conservation analysis revealed that the affected glycine residue at position 274 is highly conserved among vertebrate species, indicating significant structural or functional constraints at this site of the ACTRT1 protein (Figure 2A).

Sanger sequencing was performed for the proband and his mother, and the results confirmed an X-linked recessive inheritance pattern. The variant was hemizygous in the proband and heterozygous in his asymptomatic mother (Figure 2C). This pattern of inheritance is consistent with Mendelian inheritance for X-chromosomal disorders and suggests that the variant may be pathogenic in a hemizygous state for male patients.

The variant c.821G>C is considered likely pathogenic by pathogenicity prediction algorithms for missense variants (SIFT, SIFT4G, BLOSUM, DANN, FATHMM, PrimateAI, MetaLR, MetaSVM, DEOGEN2, EIGEN, EIGEN PC, M-CAP, Mutation assessor, MutPred, MVP, PROVEAN, LRT, and MutationTaster).

We have also created a 3D model of the mutant ACTRT1 protein and compared it with the normal protein using the Project HOPE3D tool (Figure 3).

The p.(Gly274Ala) substitution in the ACTRT1 introduces significant steric constraints due to the substantial difference in side-chain volume between glycine and alanine. The larger alanine residue cannot be accommodated within the core structure, leading to unfavorable torsion angles and steric clashes. The unique conformational flexibility of glycine is essential at this position, as it is the only residue capable of adopting the required backbone conformation without distortion. Substitution with alanine is predicted to disrupt the local protein architecture, an effect that would likely propagate to destabilize the tertiary structure of ACTRT1 and compromise its biological function. Therefore, glycine is exclusively required to maintain the native protein fold at this position.

Finally, the identified variant chrX:127185365G>C; NM_138289.3: c.821G>C, p.(Gly274Ala) in the *ACTRT1* gene was classified as a variant of uncertain clinical significance (VoUS) according to the ACMG guidelines (PM2, PP3).

## 4. Discussion

The *ACTRT1* gene encodes the actin-related protein T1, a critical structural component of a testis-specific ARP complex (ACTRT1-ACTRT2-ACTL7A-ACTL7B-ACTL9) within the PT, which is indispensable for anchoring the acrosome to the sperm nucleus and fertilization [11]. Despite its testis-restricted expression profile, the previously established OMIM phenotype associated with *ACTRT1* gene was Bazex–Dupré–Christol syndrome (BDCS; OMIM: 301845), an X-linked dominant genodermatosis characterized by follicular atrophoderma, congenital hypotrichosis, and early-onset multiple basal cell carcinomas [29,30]. However, recent evidence has definitively refuted this association. Recently, Liu et al. (2022) demonstrated that BDCS is actually caused by small intergenic tandem duplications at Xq26.1, leading to dysregulation of the *ARHGAP36* gene [31]. Importantly, no rare coding variants in the *ACTRT1* gene were identified in seven of eight BDCS families. Furthermore, immunofluorescence studies revealed the absence of *ACTRT1* expression in relevant hair follicle compartments [31]. Therefore, *ACTRT1* has been excluded as a causative gene for BDCS. Its actual phenotypic consequences remain to be fully elucidated; however, its testis-specific expression and structural role suggest potential involvement in male reproductive function, which requires further validation.

The initial link between *ACTRT1* and male infertility was established in a large-scale sequencing study. S. Chen et al. (2020) conducted a whole-exome sequencing analysis of 314 Chinese men with non-obstructive azoospermia (NOA) and severe oligozoospermia and identified *ACTRT1* as 1 of 20 novel candidate genes for male infertility [32]. Among the patients in their cohort, two patients with NOA were found to have hemizygous variants in the *ACTRT1* gene [32]. Notably, a meiotic arrest phenotype was revealed by testicular histopathology in both patients.

Subsequently, Y. Sha et al. (2021) proposed a more specific yet controversial association [33]. They identified 2 unrelated individuals with hemizygous missense variants in *ACTRT1* in a cohort of 34 infertile men with acephalic spermatozoa syndrome (ASS). The patients presented with asthenoteratozoospermia, characterized predominantly by acephalic sperm. Immunofluorescence staining revealed displaced and diffuse *ACTRT1* expression in the patients’ sperm. To validate their findings, the researchers generated an *Actrt1*-knockout mouse model using CRISPR/Cas9 and reported that approximately 60% of the sperm were headless, with TEM revealing significant defects in the head–tail coupling apparatus (HTCA). Based on this, they proposed *ACTRT1* as a gene associated with ASS. Following artificial insemination with optimized sperm, both patients and their partners achieved successful pregnancy and delivery.

However, this conclusion was challenged by a series of studies from another research group. X. Zhang et al. (2022) reanalyzed the role of *ACTRT1*, identifying it as a candidate gene in patients with syndromic teratozoospermia characterized not by headless sperm, but specifically by the detachment of the acrosome from the sperm nucleus [11]. This refined phenotype was strongly supported by functional mouse model data. Their *Actrt1*-knockout mice were severely subfertile, exhibiting deformed sperm heads and a high proportion of sperm with detached acrosomes, as confirmed by TEM. These results allowed them to reveal that ACTRT1 interacts with the ACTRT2, ACTL7A, and ACTL9 proteins to form a sperm-specific PT-ARP complex, which anchors the developing acrosome to the nucleus by connecting the inner acrosomal membrane protein SPACA1 with nuclear envelope proteins such as PARP11 and SPATA46. Importantly, their *Actrt1*-*KO* mice did not exhibit the headless sperm phenotype reported by Sha et al., despite using a similar knockout strategy. The authors argued that there was no evidence that ACTRT1 regulates the HTCA and deemed the evidence linking it to ASS as insufficient.

The human relevance of this acrosome detachment phenotype was conclusively demonstrated by Q. Zhang et al. (2024) [12]. They reported two infertile Chinese men with a ~110 kb X-chromosome microdeletion encompassing the entire *ACTRT1* gene. Semen analysis revealed sOT, and TEM confirmed sperm head deformation due to acrosome detachment, perfectly mirroring the mouse model. The deletion was inherited from the patients’ mothers, consistent with X-linked recessive inheritance. Western blot and immunostaining of patient sperm showed that ACTRT1 deficiency led to downregulated expression and ectopic distribution of ACTL7A and PLCζ, key proteins for oocyte activation, explaining the observed fertilization failure. The authors successfully overcame this challenge by using the ICSI procedure combined with artificial oocyte activation (AOA) on one patient.

In the study by H. Zhou et al. (2024), a patient with oligoasthenoteratozoospermia (OAT) was found to carry a hemizygous missense mutation in the *ACTRT1* gene [34].

The clinical and genetic features of all male patients with reported *ACTRT1* variants are summarized in Table 1.

As demonstrated by the literature, male patients with hemizygous variants in the *ACTRT1* gene exhibit a remarkably heterogeneous clinical spectrum, ranging from asthenoteratozoospermia and acephalic spermatozoa syndrome to severe oligozoospermia and non-obstructive azoospermia with meiotic arrest. This significant phenotypic variability highlights the importance of collecting and reporting detailed clinical cases that integrate comprehensive semen analysis, TEM, and genetic data. Two of the eight patients with hemizygous *ACTR1* gene variants (c.662A>G, p.(Tyr221Cys); c.431C>T, p.(Ala144Val) were diagnosed with NOA [32]. Testicular histopathology revealed meiotic arrest in both patients. Four male patients with hemizygous missense *ACTRT1* variants had oligozoospermia. At least two of these patients (both with a 110 kb deletion) and one other patient were severely oligozoospermic with sperm concentrations of 1.0, 2.0, and 0.16–2.8 × 10^6^/mL. The sperm concentration of the remaining patient was not indicated. The sperm count was normal (25.23 ± 7.62 and 28.53 ± 8.30 × 10^6^/mL) in the other patients, who were asthenoteratozoospermic men with missense *ACTRT1* variants (c.95G>A, p.(Arg32His) and c.662A>G, p.(Tyr221Cys) and ASS [33].

In our patient, a history of right inguinal cryptorchidism requiring orchiectomy could certainly be a contributing factor to his severe oligozoospermia. No other visible causes of male infertility were identified. Notably, six out of eight patients with *ACTRT1* gene variants or deletions had severe defects in spermatogenesis, such as NOA or oligozoospermia, and the cause was unknown. Furthermore, patients with *ACTRT1* deletions or missense variants exhibited severe spermatogenesis disorders. These findings suggest that *ACTRT1* deficiency may affect spermatogenesis and meiosis. Meiosis arrest was detected in patients with NOA. However, the pathogenic effect of *ACTRT1* gene variants on spermatocytes in infertile men requires further study.

Decreased sperm motility and teratozoospermia were observed in azoospermic males (Table 1). Progressive motility (PR) was very low (0–9%) in one patient. Sperm viability was also reduced in one of the four semen samples studied. However, other studies have not reported a reduced number of viable sperm cells in ejaculate. This suggests that gene abnormalities do not lead to necrospermia. The reduced motility may have been partly due to the high percentage (31%) of spermatozoa with axonemal abnormalities. Other TEM studies do not specify the number of defective gametes. However, these changes are nonspecific because they were also observed in the control group (9.7 ± 7.6%; 1–30%; see Appendix A).

Sperm head defects are the predominant morphological abnormality detected in non-azoospermic patients with *ACTRT1* gene variants or deletions [12,33,34]. Specific ultrastructural defects revealed by TEM—particularly the abnormal sperm head morphology characterized by detachment of the inner acrosomal membrane from the nucleus—are highly consistent with the phenotypes observed in both *Actrt1*-knockout mouse models and in patients with complete *ACTRT1* gene deletion [11]. Acrosomal detachment and IVF failure were also observed in two patients with a 110 kb deletion, as reported by Q. Zhang et al. [12]. This makes it difficult to determine whether acrosome detachment is a universal feature of male infertility in patients with *ACTRT1* deficiency. However, comprehensive comparisons with other reported *ACTRT1* variant cases are limited. TEM analysis was precluded for patients with NOA or severe oligozoospermia due to the absence or critically low number of spermatozoa in the ejaculate. Furthermore, data on the percentage of spermatozoa with immature chromatin or flagellar axonemal abnormalities are unavailable for other patients with identified *ACTRT1* variants. Acephalic sperm syndrome (ASS) was diagnosed in two infertile men reported by Y. Sha et al. (2021) [33]. However, the other four non-azoospermic patients did not exhibit ASS, and the role of the ACTRT1 protein in attaching the sperm head to the tail has not been demonstrated.

The significant finding in our case is the high proportion (54%) of spermatozoa with non-condensed (“immature”) chromatin. Defective chromatin compaction is a well-established factor that can impair sperm DNA integrity, potentially leading to failures in embryonic cleavage, division, and early development. Our findings provide strong circumstantial evidence linking this genetic variant to the spermatological profile, ultrastructural defects in male gametes, and failed fertilization. However, as this is a single-patient study involving a variant of unknown clinical significance, our conclusions should be interpreted with caution. Direct functional studies are required to definitively characterize the pathogenic impact of the p.(Gly274Ala) substitution on ACTRT1 function. The mechanism by which it affects acrosome formation and stability, as well as the sperm nucleus and chromatin, was not evaluated in detail. Further functional studies are necessary to definitively determine the pathogenic role of the identified variant and validate genotype-phenotype correlations.

## 5. Conclusions

This study provides a comprehensive clinical and genetic characterization of a novel *ACTRT1* gene variant in a patient with severe oligoasthenoteratozoospermia, acrosome abnormalities, and fertilization failure. The results contribute to the growing body of evidence linking *ACTRT1* deficiency to specific head defects, such as acrosomal detachment, and other sperm abnormalities, including an absence or hypoplasia of the acrosome and an absence of the postacrosomal segment. These findings support including the *ACTRT1* gene in genetic screenings for azoospermic or severe oligoasthenoteratozoospermic patients and using transmission electron microscopy for patients with teratozoospermia and fertilization failure. The phenotypic spectrum of *ACTRT1*-related infertility may extend beyond acrosomal defects to include nuclear abnormalities and other sperm morphology defects, which could contribute to the variability observed in clinical outcomes and assisted reproductive technology (ART). These efforts are essential for delineating precise genotype–phenotype correlations and understanding the full pathogenic potential of *ACTRT1* gene variants in human male infertility.

## Figures and Tables

**Figure 1 genes-16-01422-f001:**
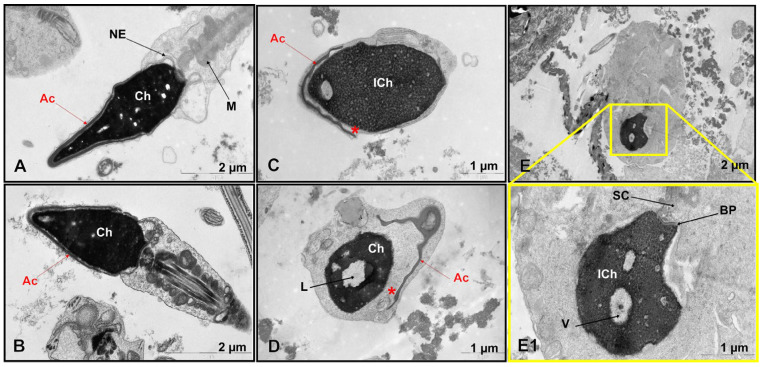
Ultrastructural sperm defects in a patient with severe oligoteratozoospermia. (**A**,**B**) Spermatozoa from a fertile control donor demonstrate normal ultrastructure, with acrosomes tightly apposed to the nucleus. (**C**–**E**,**E1**) Spermatozoa from the patient showing characteristic pathological features, including acrosomal hypoplasia and detachment from the nuclear envelope (**C**,**D**), acrosomal aplasia (**E**,**E1**), and impaired chromatin condensation (**C**,**E1**). Abbreviations: Ac: acrosome (red arrows); Ch: chromatin; NE: nuclear envelope; M: mitochondria; ICh: immature chromatin; L: lacuna; *: the enlarged space between the acrosome and the nucleus; V: vacuole; BP: basal plate of the sperm neck; SC: segmented columns of the connecting piece of the sperm flagellum.

**Figure 2 genes-16-01422-f002:**
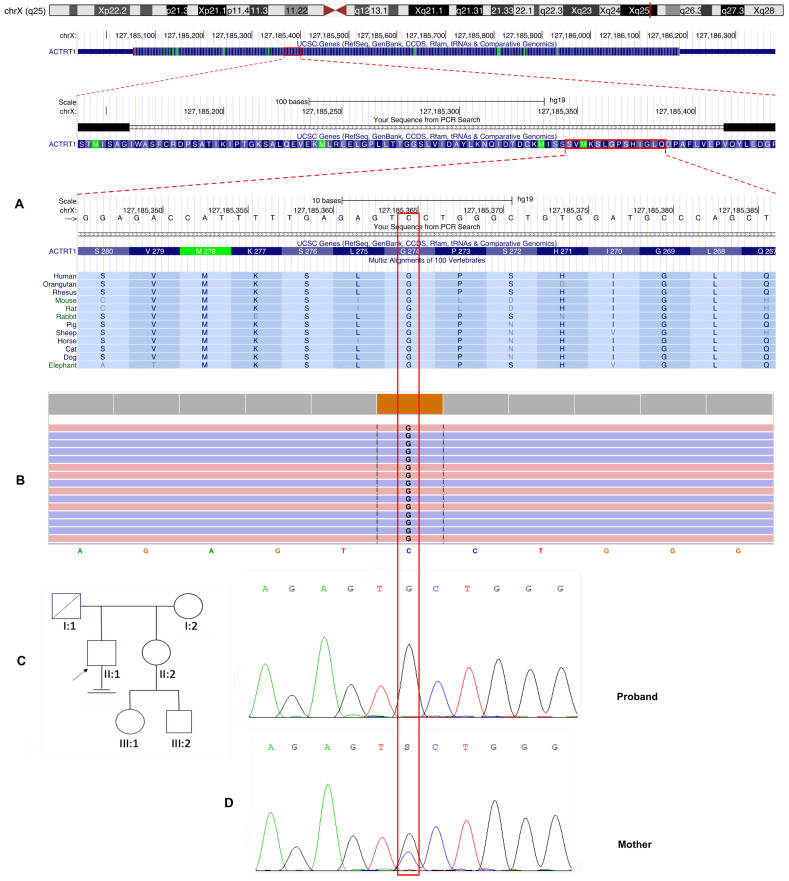
Identification and validation of a novel *ACTRT1* gene variant. (**A**) Schematic localization of the X-linked *ACTRT1* gene and evolutionary conservation analysis. The gene structure comprises a single exon. The identified c.821G>C, p.(Gly274Ala) missense variant is indicated at the genomic coordinate chrX:127185365. Multiple sequence alignment across diverse species demonstrates that the glycine residue at position 274 is completely evolutionarily conserved (indicated by a red frame), highlighting its putative structural and functional importance. (**B**) Visualization of the hemizygous chrX:127185365C>G substitution in the *ACTRT1* gene using Integrative Genomics Viewer (IGV v2.17.4) software. (**C**) Pedigree of a patient with sOAT. The proband is indicated by an arrow. Circles represent females, and squares represent males. Deceased family members are denoted by a diagonal slash. A vertical line with two horizontal crossbars marks the infertile couple. (**D**) Sanger sequencing of a proband and his mother. Sanger sequencing confirmed the c.821G>C, p.(Gly274Ala) substitution in a hemizygous state in the proband and in a heterozygous state in the maternal carrier. The substitution site is highlighted with a red frame.

**Figure 3 genes-16-01422-f003:**
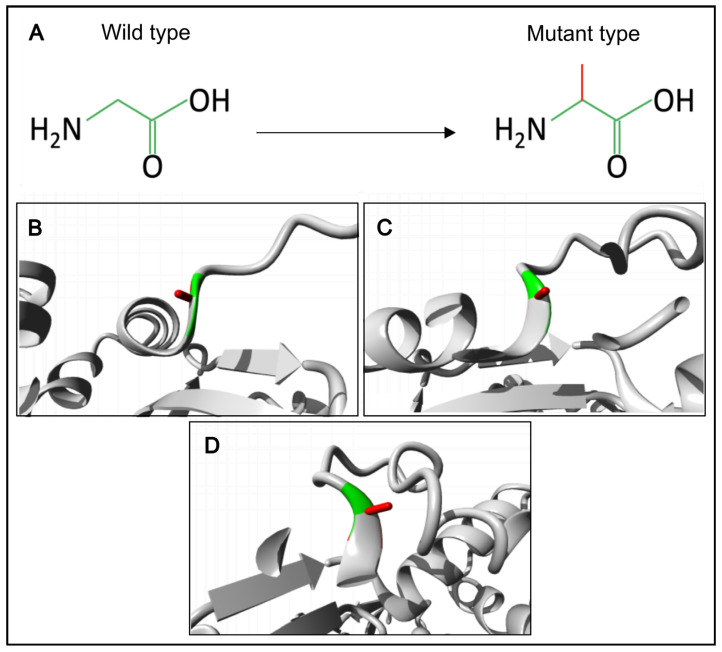
Three-dimensional model of normal and mutant ACTRT1 protein structure. (**A**) A schematic comparison of the wild-type glycine residue (left) and the mutant alanine residue (right), with the common backbone shown in green and the unique side chains shown in red. (**B**–**D**) Close-up views of the substitution site from different angles, with the protein backbone shown in gray and the wild-type and mutant side chains shown in green and red, respectively.

**Table 1 genes-16-01422-t001:** Clinical and genetic characteristics of 8 infertile male patients with *ACTRT1* (NM_138289.4) gene variants/deletions.

Patient	Gene Variant	Phenotype/Additional Data	References
NOA54	c.662A>G, p.(Tyr221Cys)	NOA; meiotic arrest. Normal male karyotype (46, XY), no AZF microdeletions. Normal hormones (FSH, LH, testosterone) and testicular volume	S. Chen et al., 2020 [32]
NOA281	c.431C>T, p.(Ala144Val)		
F018	c.95G>A, p.(Arg32His)	AT; ASS	Y. Sha et al., 2021 [33]
F034	c.662A>G, p.(Tyr221Cys)		
L053	110-kb deletion	sOT; acrosome detachment; fertilization failure	Q. Zhang et al., 2024 [12]
L116			
M1555	c.169G>A, p.(Val157Met)	OAT. Normal male karyotype (46, XY), no AZF microdeletions	H. Zhou et al., 2024 [34]
Ex2862	c.821G>C, p.(Gly274Ala)	sOAT; acrosome detachment; fertilization failure	Present study

NOA—non-obstructive azoospermia; FSH—follicle-stimulating hormone; LH—luteinizing hormone; ASS—acephalic sperm syndrome; AT—asthenoteratozoospermia; sOT—severe oligoteratozoospermia; sOAT—severe oligoteratozoospermia.

## Data Availability

The original contributions presented in the study are included in the article/Appendix A. Further inquiries can be directed to the corresponding author.

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
