# Peer review of "A Novel Variant of the ACTRT1 Gene Is Potentially Associated with Oligoasthenoteratozoospermia, Acrosome Detachment, and Fertilization Failure"

_genes, 2025, doi:10.3390/genes16121422_

Round 1
Reviewer 1 Report
Comments and Suggestions for Authors
In this case report, Solovova et al. perform a comprehensive evaluation of a patient with primary infertility and severe oligoasthenoteratozoospermia. Transmission Electron Microscopy revealed significant ultrastructural abnormalities, particularly acrosome detachment in 49% of spermatozoa and a high percentage (54%) of cells with insufficiently condensed chromatin. Whole Exome Sequencing identified a novel hemizygous missense variant, c.821G>C, in the ACTRT1 gene. Although this variant was classified as a Variant of Uncertain Significance, in silico analyses and 3D protein modeling indicated that the mutation is likely deleterious, as it disrupts the ACTRT1 protein. The authors therefore propose that this novel ACTRT1 mutation is associated with the specific teratozoospermia phenotype that is characterized by acrosome detachment and fertilization failure, supporting the importance of genetic testing and sperm ultrastructural analysis for infertile men with specific morphological abnormalities.
General comments:
Overall, the manuscript is well written, provides a thorough literature review, and highlights important genetic contributors to sperm acrosome abnormalities.
- The authors characterize this mutation as a variant of unknown significance but refer to it using language that implies causality. Although in silico analyses were performed, this is not sufficient to consider this mutation as causative, as it cannot provide conclusive evidence. The authors may consider gene editing experiments or introducing the mutation in an in vitro or in vivo model system to obtain more robust functional evidence. Otherwise, we suggest that the authors avoid using causal language claiming that the variant “can cause” teratozoospermia or that the study “demonstrates” an effect. Instead, we recommend revising the manuscript to state that the mutation is “potentially associated”, “consistent with” or “may suggest” an involvement in teratozoospermia.
- Assays for functional validation, such as immunofluorescence for ACTRT1 protein expression, PLCζ localization, and acrosome staining, should be included.
- Did the authors observe any spermatozoa with completely round heads? If confirmed, that would be a valuable contribution to the list of genes that are currently known to contribute to spermiogenesis, the development of the acrosome (SPATA, PLK4), and are associated with round-headed spermatozoa and therefore in chromatin condensation. If so, a discussion on potential issues with chromatin compaction, and data from the aniline blue assay, should be included in the manuscript.
- We suggest including a more detailed description of the sperm acrosomal defects that were identified in this patient. Again, acrosome staining would be helpful to determine whether the acrosome was actually missing. If so, what proportion of spermatozoa were affected and what was the fertilization outcome? A discussion of these aspects, if possible, would yield valuable biological proof and strengthen the manuscript.
Minor Comments:
- If possible, include the sequencing depth and uniformity metrics of the ACTRT1 region, specifically.
Reviewer 2 Report
Comments and Suggestions for Authors
In this case report, the authors describe the comprehensive clinical and genetic evaluation of a male patient with severe oligoteratozoospermia and primary infertility, identifying a novel hemizygous missense variant in the ACTRT1 gene that may be responsible for a specific form of X-linked teratozoospermia characterized by acrosome detachment from the sperm nucleus.
As this is a single case report, the study is inherently limited to specific observations from one individual. This naturally restricts the generalizability and overall impact of the findings, although the identification of a novel ACTRT1 variant and its potential pathogenic relevance represent interesting and valuable contributions to the field of male infertility genetics.
- It would be important to clarify why the authors used the WHO Laboratory Manual for the Examination and Processing of Human Semen, 5th edition (2010), instead of the 6th edition (2021), which provides the most updated reference standards for semen analysis.
- The discussion section could be better organized to more clearly integrate the presented case with the existing body of literature. A more systematic comparison between the patient’s clinical, ultrastructural, and genetic findings and those previously reported for ACTRT1 variants would strengthen the interpretation and help clarify how this case fits within the broader phenotypic spectrum of ACTRT1-related male infertility.
- Finally, given that this is a single-patient study, the conclusions should be slightly more cautious and emphasize the need for further functional and clinical studies to confirm the pathogenic role of the identified variant.
Reviewer 3 Report
Comments and Suggestions for Authors
The manuscript “A novel variant of the ACTRT1 gene is potentially associated with oligoasthenoteratozoospermia, acrosome detachment, and fertilization failure” provides information on the effects of a variant of the ACTRT1 gene on sperm formation. Personally, I believe that basic results of this type can contribute to the application of clinical research in the future. The results in this study are very interesting, and if the results can be applied clinically in the future, it could make a significant contribution to ART. Also, it needs to add the schemes of conclusion to make it easier for readers to understand.
- Is it known why only one copy of the variant occurs? Are external factors (especially lifestyle), environment, nutrition, etc, involved? Also, is it possible that the region of residence or race may make the mutation more likely?
- According to the phenotype of sperm in a variant of the ACTRT1 gene, natural fertilization seems impossible, but is fertilization possible with ICSI?
- In this study, the data of transmission electron microscopy showed acrosome detachment from the nucleus in 49% of the sperm. It means 51% of the sperm are normal? Even if there is a variant, are half of the sperm normal?
- We understand the meaning of hemizygous, but I think adding an illustration would make it easier to understand. Also, summarizing the results in a scheme will improve understanding.
Round 2
Reviewer 2 Report
Comments and Suggestions for Authors
Authors revised the manuscript according my comments.